

# Enhancing resilience to coastal flooding from severe storms in the USA: International lessons

Darren M. Lumbroso[1], Natalie R. Suckall[2], Robert J. Nicholls[3], Kathleen D. White[4]

[1]HR Wallingford, Howbery Park, Wallingford, Oxfordshire OX10 8BA, UK
[2]Geography and Environment, University of Southampton, University Road, Southampton SO17 1BJ, UK
[3]Engineering and Environment, University of Southampton, University Road, Southampton SO17 1BJ, UK
[4]US Army Corps of Engineers, Institute for Water Resources, National Capital Region (NCR), 7701 Telegraph Road (Casey Building), Alexandria, Virginia 22315, USA

*Correspondence to*: D. M. Lumbroso (d.lumbroso@hrwallingford.com)

**Abstract.** Recent events in the USA have highlighted a lack of resilience in the coastal population to coastal flooding, especially amongst disadvantaged and isolated communities. Some low income countries, such as Cuba and Bangladesh, have made significant progress towards transformed societies that are more resilient to the impacts of cyclones and coastal flooding. To understand how this has come about a systematic review of the peer-reviewed and grey literature related to resilience of communities to coastal flooding was undertaken in both countries. In both Cuba and Bangladesh the trust
between national and local authorities, community leaders and civil society is high. As a consequence evacuation warnings are generally followed and communities are well prepared. As a result over the past 25 years in Bangladesh the number of deaths directly related to cyclones and coastal flooding has decreased, despite an increase of almost 50% in the number of people exposed to these hazards. In Cuba, over the course of eight hurricanes between 2003 and 2011, the normalised number of deaths related to cyclones and coastal floods was an order of magnitude less than in the USA. In low-income
countries, warning systems and effective shelter/evacuation systems, combined with high levels of disaster risk reduction education and social cohesion, coupled with trust between government authorities and vulnerable communities can help to increase resilience to coastal hazards and tropical cyclones. In the USA, transferable lessons include improving communication and the awareness of the risk posed by coastal surges, mainstreaming disaster risk reduction into the education system and building trusted community networks to help isolated and disadvantaged communities, and improve
community's resilience.

## 1   Introduction

People living in coastal areas face increasing storm-induced hazards of flooding caused by changing sea levels, high winds and waves, especially during tropical storms (Kron, 2013; Wong et al., 2014). As a result coastal populations are faced with property damages, economic losses, and in the worst cases, loss of life (Nicholls, 2006). Both high- and low-income
countries are affected by tropical storms. Typhoon Haiyan (2013) in the Philippines, resulted in at least 6,300 deaths and





more than US$2 billion in damages (Lagmay et al., 2015), whilst Hurricane Katrina (2005) in the USA resulted in some 1,800 deaths and US$50 billion in losses (Kates et al., 2006).

Over the past 60 years, the disparity between the rich and poor has increased in the USA, and the poorest people, who tend to live in communities on the fringes of urban areas, in inner cities and remote rural areas, have become increasing
physically and socially isolated (Morrow, 1999; Reardon, 2011). These isolated households often have limited social networks to call upon in times of emergency, making them more vulnerable and less resilient to natural hazards than better connected households (Morrow, 1999; Srinivasan et al., 2003; Macardle, 2014).

Currently, almost 3% of the USA's population lives in areas directly subject to a 1% annual exceedance probability coastal flood (Crowell et al., 2013). Over the past decade, coastal flooding resulting from extreme events, such as Hurricanes
Katrina, Rita and Sandy, has highlighted a lack of resilience in the coastal population of the USA, especially amongst disadvantaged and isolated communities. The 13-state region of the US South-East is particularly vulnerable in that it contains approximately 80% of all the US counties that experience persistent poverty[1] (Thompson and Gaviria, 2004). There has also been an increase in vulnerability as a result of more people in hurricane affected coastal counties living in mobile homes. Between 1970 and 2000, the percentage of the total population living in mobile homes in coastal counties increased
from 8.4% to 23.2% (Cutter et al., 2007). Coastal flooding may be further exacerbated by climate change, including changing sea-levels. Of 52 communities located on the USA's East and Gulf Coasts, two-thirds are likely to experience a tripling or more in the number of annual high-tide floods by 2030 (Spanger-Siegfried et al., 2014). This is a particular problem in the Gulf region of the US, where over 99% of the most socially vulnerable people[2] live in areas likely to be subject to coastal flooding, compared to only 8% of the least socially vulnerable (Martinich et al., 2013).

The future impact of climate change on coastal flooding, the combination of growing coastal populations and rising sea levels, creates a major hazard that requires urgent adaptation. The impacts of coastal flooding can be large, and some measures that reduce losses can be highly cost effective. In the USA, one study of spending on relief and recovery found that the federal government spent US$136 billion on disaster relief between 2011 and 2013; the equivalent of nearly US$400 per household per year (Weiss and Weidman, 2013). There is a need to establish new and improved policies, strategies and
measures capable of adapting to an uncertain future that encompass integrated portfolios of structural and non-structural measures. In this paper structural measures are considered to involve any physical infrastructure to reduce or avoid possible impacts of floods (e.g., flood defences and barriers). Non-structural measures are those that do not require infrastructure, such as policies and laws, improved warnings, public awareness raising, training and education, as well as flood proofing or elevating buildings, relocation, and floodplain management. For an integrated portfolio of measures to be most successful,
there is a need to increase the resilience of communities, especially disadvantaged ones.

---

[1] This is a county in the USA where 20% or more of its population has lived in poverty over the past 30 years based on US census data
[2] Where social vulnerability is defined by the Social Vulnerability Index (SoVI), originally formulated by Cutter et al. (2003)



To examine how the USA may be able to increase its resilience to coastal floods, we have considered experience from low-income countries. Here, governments face significant financial constraints in providing infrastructure to protect populations from severe storms. Despite financial challenges, some low-income countries have made significant progress in reducing deaths, as well as ensuring that people are able to maintain their livelihoods following events, by focusing on

increasing resilience. These approaches do not depend on hard and soft infrastructure alone but also include interventions that draw on and enhance social capital[3] and self-sufficiency of the community (Adger, 2003, Norris et al., 2008, Cash et al., 2013). These successes present an opportunity for cross-country learning on how to increase the resilience of their own disadvantaged communities to coastal flooding.

In this paper, we aim to analyse how the resilience of communities has been increased in Bangladesh and Cuba, including

this social dimension, and consider the lessons for the USA. These countries were investigated because they both experience severe storms that can significantly affect large and vulnerable coastal populations, and both countries have made significant progress towards transformed societies that are more resilient to the impacts of cyclones and coastal flooding. To understand the transferable lessons from Bangladesh and Cuba, we conducted a systematic review of the literature on resilience of communities to coastal storms and surges.

The first part of the paper details the definition of resilience used in this paper and provides background American communities' resilience to coastal flooding. The second part of the paper provides background on the case study countries before introducing the systematic review method used in this research. The measures that have been used to increase communities' resilience to coastal flooding and cyclones in Bangladesh and Cuba are then discussed, together with lessons that can be drawn from these for the USA.

## 20   2   Resilience: absorbing, adapting and transforming

### 2.1 Conceptual background

In the context of this paper, we use the United States Army Corps of Engineers' (USACE) definition of resilience, which is "*the ability to anticipate, prepare for, and adapt to changing conditions and withstand, respond to, and recover rapidly from disruptions*" (Schultz and Smith, 2016). In our conceptualisation of resilience, we recognise the importance of the

political, social and cultural relationships that underpin society and its capacity to respond to stress. Within that context, we consider Béné, et al.'s (2012) concept of resilience comprising three components: (1) absorption, (2) adaptation and (3) transformation as follows:

1.      **Absorption** is concerned with resisting the initial impacts of a shock. It relies on strategies to cope with the shock rather than instigating a significant change to reduce or avoid future shocks.

---

[3] Social capital can be features of social organization such as networks, norms, and social trust that facilitate coordination and cooperation for mutual benefit (Putnam, 1995)





2.    **Adaptation** refers to the reactive or anticipatory responses that reduce risk and enable households, communities or governments to continue to function during and after the shock, but without significant structural changes.

3.    **Transformation** refers to changes that alter societal functioning to a new and fundamentally improved state (Walker et al., 2004, Folke et al., 2010). This may occur as the cumulative effect of sustained incremental adaptations (Park et al., 2012, Kates et al., 2012) or it can also take place more rapidly.  Transformation may be reactive or anticipated. Where transformation occurs it demands a substantial change away from the status quo (Lonsdale et al, 2015).

All three of these components are generally present where communities have increased their resilience to natural hazards. Fig. 1 illustrates the relationship between adsorption, adaptation, transformation and resilience. External shocks such as floods which severely impact communities can lead to changes in policies and the implementation of measures that can help to increase communities' resilience to future events (Birkland, 2006).

## 2.1 Resilience of communities in the USA to coastal flooding

Over the past decade or so, coastal flooding from severe storms in the USA, most especially Hurricanes Katrina and Sandy, has disproportionately affected disadvantaged communities including low income home owners (Elliott and Pais, 2006); the inner-city poor (Cutter et al., 2006); older people (Brunkard et al., 2008); and minorities (Dyson, 2006). Katrina saw a significant loss of life, and the mortality rate was up to four times higher among non-whites compared to whites (Brunkard et al., 2008).  Furthermore, whilst around 80% of the population left New Orleans in advance of Hurricane Katrina and escaped the risk of death during the storm, the socially disadvantaged who lacked access to transport, money for fuel, or alternative accommodation were forced to stay in their homes or evacuate locally (Brodie et al., 2006). In Hurricane Sandy, the majority of the people who died were older residents (over 60) (CDC, 2013).  Of the 20 people who drowned in their homes in New York, a major reason for not evacuating was an inability to leave owing to a lack of transportation (CDC, 2013).

With respect to economic status, disadvantaged groups are also disproportionately affected. For example, 12 months after Katrina, the unemployment rate amongst non-returnees to an affected state was 35.7%, compared to a national average of 3.7% (Zissimopoulos and Karoly, 2010). Non-returnees were more likely to be young, unmarried, black women who did not have a university degree. In terms of property loss, 53% of black residents reported they lost everything, compared with only 19% of white residents (Lavelle, 2006). Similarly, following Sandy the majority of those who registered for Federal Emergency Management Agency (FEMA) assistance following New York City's storm surge were renters with low incomes (FEMA, 2010).

Following Hurricane Sandy, a range of coastal risk reduction measures were considered for New York including a large surge barrier, although more limited measures were initially selected (USACE, 2015). For example, around lower Manhattan multi-functional flood defences and measures to flood-proof buildings are being implemented. However, isolated and socially disadvantaged communities often remain vulnerable to coastal flooding.  In some parts of the USA, there has not



been an extreme coastal surge for many years, and as a result, awareness of the risk posed by coastal surges is low. For example, the state of Georgia has not experienced an extreme surge for more than 100 years. A recent survey found that less than one quarter of Georgians living on the coast were very concerned about hurricanes and many people living in low lying areas did not believe their homes could flood (USACE, 2014). The last significant hurricane landfall in Glynn County, Georgia was in 1898, which caused a 5 m high storm surge in the town of Brunswick. It has been estimated that should a similar sized coastal surge occur in Brunswick today it would result in fatality rates of around 45% (Lumbroso et al., 2015).

## 3 Case study countries: Bangladesh and Cuba

### 3.1 Bangladesh

Bangladesh is located in South Asia, on the delta of the two largest rivers on the Indian subcontinent, the Ganges and Brahmaputra. It covers 144,000 km$^2$, (slightly larger than the state of New York), with large areas less than one metre above high tide (Huq et al., 1995; Agrawala et al., 2003). Of the approximately 160 million population, around 50% live in the coastal floodplain (Neumann et al., 2015). Bangladesh is a low income country; in a recent analysis based on Gross Domestic Product (GDP) per capita adjusted for purchasing power parity, it ranks 140[th] out of 185 countries (Pasquali, 2015).

Bangladesh is especially vulnerable to cyclones because of its location at the triangular shaped head of the Bay of Bengal, the low lying nature of its coastal area, its high population density and the dearth of significant coastal flood defences (Haque et al., 2012). It is rated as one of the world's most vulnerable countries to tropical cyclones (UNDP, 2004). There has been at least 20 severe cyclonic storms making landfall since 1970 (Paul, 2009). In November 1970 Cyclone Bhola killed as many as 500,000 people (Paul, 2010) and in April 1991 Cyclone Gorky, accompanied by a tidal surge of up to 10 m, resulted in about 140,000 deaths (Chowdhury et al., 1993). Fig. 2 shows the number of cyclone-related fatalities and population growth between 1960 and 2015.

### 3.2 Cuba

Cuba is a large Caribbean island nation. The 2012 census estimated Cuba's population to be 11.1 million, with a population density of 100.7 inhabitants per square kilometre. Based on GDP per capita adjusted for purchasing power parity it is ranked 136th in the world (CIA, 2016) and is a low income country. Hurricanes are the major natural hazard to affect Cuba. In 1926 a hurricane caused 600 deaths and in November 1932 a hurricane with wind speeds of almost 250 km/hour caused a storm surge 6 m high and led to some 3,000 deaths (Cubahuriccanes.org, 2016). In 1963 Hurricane Flora resulted in some 1,500 deaths (Pichler & Striessnig, 2013). The island is affected by the same hurricane systems that impact the Eastern and Gulf coasts of the US. However, over the course of eight hurricanes between 2003 and 2011, there were 44.73 deaths per million people at risk in the USA and only 2.43 deaths per million people at risk in Cuba (Newhouse, 2011).



## 4 Systematic review method

To develop an understanding of how the resilience of communities to coastal flooding and tropical cyclones in Bangladesh and Cuba has increased, we conducted a systematic review of the peer-reviewed and grey literature in both countries. With origins in the health science literature, systematic reviews are increasingly used to explore climate change
adaptation (Berrang-Ford et al., 2011, Antwi-Agyei et al., 2014, Porter et al., 2014). An explicit procedure is followed whereby documents that focus on a specific issue are analysed according to clearly formulated questions. In this review, we aimed to establish what measures have contributed towards increasing community' resilience in Bangladesh and Cuba?
We follow a realist approach (see Pawson et al., 2005). This approach is more suited to understanding why and how complex social interventions work (or fail) rather than stating the outcome of, for example, medical interventions as in the case in
Cochrane-style reviews, which often aim to quantify the effectiveness of a discrete intervention (Pawson and Tilley, 2004). In using such an approach, we focus on a deep qualitative understanding rather than a broad understanding of the problem. As such, we apply a strict inclusion and exclusion criteria, shown in Table 1, and fewer documents are considered than in other review approaches (Thompson et al., 2010).

We followed a two-step process in retrieving articles for potential review. Firstly, to identify articles of interest in the
peer-reviewed literature, we conducted a keyword search in the ISI Web of Science. We used the words "Climat*," "Adapt*," "Cyclon*," "Hurricane" , "Flood*," "Resili*," or "Storm-surge" coupled with the term "Bangladesh" or "Cuba". Each abstract was reviewed and articles that were irrelevant were excluded based on the criteria outlined in Table 1. Second, to identify articles in the grey-literature, including Non-Governmental Organisation (NGO) documents, working papers and university theses, we used a snowballing method whereby advice on relevant publications was sought from key experts
(Hagen-Zanker and Mallett, 2013). We recognise that this method may create a researcher bias because experts may not be objective; however, we also recognise that this method can highlight important information not represented in peer-reviewed journal articles (Hagen-Zanker and Mallett, 2013).

Overall, we excluded 1518 peer reviewed documents from the review for the reasons presented in Table 2. We reviewed 43 documents, shown in Table 2. This comprised 76 papers on Bangladesh:  25 peer reviewed and 51 grey literature
documents. From Cuba, we review six peer-reviewed and 45 grey literature documents.  In the following section, we present results of our analysis.

## 5 Results of the systematic review

### 5.1 Increasing communities' resilience to coastal flooding and cyclones in Bangladesh

Over the past 55 years a number of coastal flood risk reduction measures have been implemented in Bangladesh.
However, it is only in the past 25 years that these have had an impact in significantly reducing fatalities from cyclones and coastal flooding, despite an increase in the number of people exposed (see Fig. 2). This decrease in fatalities could arguably



be a measure of the improvement in community resilience. Fig. 3 shows the number of deaths caused by cyclones and the risk reduction measures that have been implemented that have increased the population's resilience to coastal floods. This section outlines the measures that have been employed over the past 55 years in Bangladesh and the impact that they have had on increasing community resilience.

### 5.1.1 Improved preparedness and risk communication

One of the keys to increasing community resilience and reducing loss of life is improving preparedness and the ability to take precautionary measures in response to threats. In 1965 the idea of a Cyclone Preparedness Programme (CPP) was introduced when the National Red Cross Society, now the Bangladesh Red Crescent Society, asked the International Federation of the Red Cross and Crescent (IFRC) to support the establishment of an early warning system for the people
living in the coastal belt of Bangladesh (Mathbor, 2007). Following independence, the CPP was officially formed in 1972 by the Government of Bangladesh and the Bangladesh Red Crescent Society. The CPP's main vision is to reduce the risk and loss of human life and also to reduce the economic losses from the cyclones and tsunamis, mainly for the vulnerable communities living in the coastal belt of Bangladesh (Ha and Ahmed, 2015). CPP volunteers are trained in early warning dissemination, evacuation, search and rescue, first aid and relief operations. Basic disaster management and leadership
training is also part of their capacity development process.

By the 2000 the CPP had 32,000 volunteers, which has risen to 50,0000 volunteers in 2015 (Ha and Ahmed, 2015). Volunteers serve without monetary benefit and are motivated by altruistic concerns for their family, friends and community, social responsibility, religious benefits, reputation and desire to gain community recognition (Amin, 2012). Past experience of cyclones in 1970, 1991 and 2007 has also been found to provide motivation for volunteers (Amin, 2012).
The CPP can now alert around eight million people across the whole coastal region, and can assist around four million of these to evacuate. The warning system uses Asia's largest radio network, linking the CPP's Dhaka headquarters with 143 radio stations. Radio warnings are then relayed by village-based volunteers using megaphones and hand operated sirens. The volunteers are also trained to rescue people and evacuate them to shelters, administer first aid, and assist in post-cyclone damage assessment and relief (Majumder, 2013).
Preparedness at a national government level has also improved. In 1985, the Bangladesh government introduced a Standing Orders for Cyclones document. This document outlines specific directives, duties, and responsibilities regarding disaster management for all relevant public agencies, including regional and local administrators, relevant ministries, and the Bangladesh Army, Navy and Air Force (Government of Bangladesh, 1999). These documents were updated in 2010 (WMO, 2011).
In 1997, the Bangladesh Red Crescent Society launched a comprehensive Community-Based Disaster Preparedness programme with the aim of increasing the resilience of communities living in high-risk areas of disaster-prone districts (Bangladesh Red Crescent Society, 2008). This community-based programme focuses particularly on women, as well as helping to increase community awareness concerning risks and how they can collectively act to reduce their exposure to



hazards (Bangladesh Red Crescent Society, 2008). It also fosters community participation and unity to coordinate with the local government in fulfilling their responsibilities to save lives (Bangladesh Red Crescent Society, 2008). The programme assists communities to organise themselves in "micro-groups" with 30 to 40 members in each, which act as the target group for household-level disaster risk reduction interventions (Bangladesh Red Crescent Society, 2008). As part of the

programme each community has volunteers who act as Community Disaster Response Teams and are trained to help vulnerable community members during extreme events.

There have also been efforts in Bangladesh to introduce disaster risk reduction into the education system. A learning kit for children on disaster risk reduction was developed and adapted to local contexts and language. The learning kit developed in 2005 was the first material of this type in the Bangla language aimed at helping children learn about disaster risk and take

actions for risk reduction (UNISDR, 2007).

### 5.1.2 Cyclone shelters

The CPP's success has not only been as a result of improving awareness, preparedness and the dissemination of warnings, but also as a result of an improvement in the provision of cyclone shelters, together with the forecasting and planning for future cyclones. As early as the 1960s, some limited structural measures were being implemented to increase

communities' resilience to coastal floods and tropical cyclones. These primarily took the form of purpose-built cyclone shelters, which were constructed in the 1960s for local vertical evacuation purposes only. Horizontal evacuation is not feasible owing to the large numbers of people, the large distances involved and the poor transport systems in Bangladesh. The shelters usually comprise concrete buildings raised on stilts or earth mounds so that they are raised above the extreme coastal surge level, which is often 4 m or more above the ground level. However, in the 1960s, there were only 100 cyclone

shelters in the country (Rahman and Islam, 2011) hardly sufficient for multi-million population of the coastal floodplain.

After Cyclone Bhola in 1970, which cause half a million deaths (Figs. 2 and 3), some 200 new cyclone shelters were constructed in the coastal areas of Bangladesh. Given that the average capacity of a shelter has been estimated to be around 1,600 people (Dasgupta et al., 2010), these new structures did little to increase communities' resilience to coastal surges. Most of these original shelters are now unusable owing to structural issues or geomorphological changes in the coastal

landscape (Jia, 2012).

After the 1991 cyclone, which resulted in 140,000 fatalities, a Task Force on Cyclone Shelters was set up by the Planning Commission of Bangladesh. A Multipurpose Cyclone Shelters Programme was commissioned and the planning and design of new shelters were comprehensively studied by the Government of Bangladesh, UNDP, the World Bank (Jia, 2012; Habib et al., 2012). This led to some 2,000 multipurpose cyclone shelters being constructed. Most of these multipurpose cyclone

shelters have been designed for use on a day-to-day basis as primary schools typically with 250 students (Jia, 2012). Primary level education in coastal areas of Bangladesh is closely linked with cyclone shelters. The reasoning behind multipurpose shelters is that this allows for their continued use and upkeep in the periods between cyclones and also provides educational co-benefits by providing a location for primary schools in the coastal districts of the country.



Lessons learned over the past decades in the construction of cyclone shelters have further improved their designs. For example, shelters now include separate spaces for women, as the result of studies showing that women often did not use shelters because separate accommodation and washing facilities had not been provided for. In some cases, cyclone shelters were felt by women to be the domain of men, because when they were not being used as shelters they served as mosques or

madrasahs[4] (Hafiza and Neelormi, 2015). Many women believed that they were not sanctioned to enter such buildings (Hafiza and Neelormi, 2015).

Livestock plays a major role in Bangladeshi's lives, and their loss in a disaster can have a significant negative effect on income. In newer shelters, space for livestock is now standard in the designs. This feature helps communities to improved economic recovery following coastal flooding. However, there is still much work to be done, as it has been estimated that

currently only around 2% of all cyclone shelters in Bangladesh have any kind of livestock facility (IRIN, 2012).

The construction of cyclone shelters is part of an ongoing programme. It has been estimated that by 2020 about 4,760 new shelters are required in Bangladesh's 14 coastal districts where the CPP is operational and 7,124 by the year 2025 (World Bank, 2014). The construction of these shelters will be phased, with priority based on the population density, coastal surge level, distance from existing shelters and the needs of the stakeholders (World Bank, 2014).

**5.1.3 Improved forecasting skill**

Together with the formation of the CPP, construction of cyclone shelters and the dissemination of warnings; improvements in the accuracy of cyclone forecasts has also played a part in increasing resilience. The Bangladesh Meteorological Department is the organisation responsible for forecasting and warning for cyclones in Bangladesh (Miyan, 2005). In 1988 two S-band radars were implemented on the coast to improve cyclone forecasting, and in 2009 these were

upgraded with Doppler radar systems. The Cyclone Warming Programme is managed by the Storm Warning Centre of Bangladesh Meteorological Department. It monitors the cyclonic disturbances in the Bay of Bengal and advises the Government of Bangladesh on warnings that are issued in two stages. The first is an alert stage that provides a warning at least 36 hours ahead of formation of cyclonic depression in the Bay of Bengal. The warning stage has three levels: "Warning" 24 hours before landfall; "Danger" at least 18 hours before landfall and "Great danger" at least 10 hours before

landfall (WMO, 2011).

**5.1.4 Implications**

The construction of multi-purpose shelters, together with improvements in cyclone forecasting, communication of early warnings, education, widespread mobilisation of volunteers and community preparedness meant that the number of fatalities as a result of Cyclone Sidr were over 30 times lower than the cyclone of 1991, despite Cylone Sidr being of the same

strength and the population at risk having increased by 37% (Paul, 2010). It is estimated that some 1.5 million people took

---

[4] Institutions for the study of Islamic theology and religious law



refuge in cyclone shelters during Cyclone Sidr in 2007 (Ministry of Environment and Forests, 2009). Trust in governments and institutions such as the CPP during emergencies is generally very high. Survey data from two Bangladeshi islands show that around 98% of residents have trust in the early warning system (Paul and Rahman, 2006). More recently a survey of 200 households in coastal areas affected by Cyclone Sidr in 2007 and the less severe Cyclone Mahasen in 2013 found that 90.5%

of the 200 respondents received warnings during Sidr, and this number increased to 96% during Mahasen (Roy and Kovordanyi, 2015; Roy, 2016). It also found that 83% of the respondents claimed the warnings to be understandable (Roy, 2016). This trust relates to the accuracy of the message (i.e. that a storm will follow) and confidence in local evacuation plans.

As Fig. 3 shows there have been a number of adaptation measures implemented in Bangladesh over the past 50 years that

have increased resilience of communities in terms of increasing their absorptive, adaptive and transformative capacities. Fig. 4 shows how resilience has developed in Bangladesh over the past 50 years in terms of adsorption, adaptation and transformation components. It has been a combination of measures that have led to a transformation in the way communities' function in the face of cyclones and coastal flooding. In Bangladesh this transformation has been the result of incremental increases in resilience. There has been a shift from a relief centric approach to disaster risk reduction to a more

holistic, multi-disciplinary disaster risk reduction approach.

The first cyclone shelters constructed in the 1960s did little to increase communities' resilience owing to their limited number; deficiencies in their design; accuracy of the cyclone forecasts and the limited channels via which warnings were disseminated. It has generally been major events (i.e. Cyclones Bjola, Gorky and to a lesser degree Sidr) that have acted as catalysts for the implementation of adaptation measures to increase resilience. Fig. 2 shows that despite the number of

people in Bangladesh increasing by 250% over the past 55 years since 1991 the number of fatalities has decreased significantly. However, with sea-level rise and deltaic subsidence increasing flood potential and a population growth rate of 1.2%, the process of transformation is ongoing and much remains to be done. The current cyclone shelters can only provide refuges for around 50% of the targeted population and are unevenly distributed in the areas at risk from cyclones and coastal flooding. Another issue is that there is often little funding for maintenance of shelters after they have been completed. If the

goal of constructing some additional 7,000 cyclone shelters over the next decade is not achieved it is possible that the increase resilience in Bangladesh may slow down.

**5.2 Increasing communities' resilience to coastal flooding and cyclones in Cuba**

Fig. 5 shows a timeline of the hurricanes that have affected Cuba since 1960, the fatalities attributed to hurricanes each decade and the measures that have been implemented that have increased the population's resilience. This sections outlines

the different measures that have increased communities' resilience to hurricanes over the past 50 years.



### 5.2.1 Improved forecasting skill and credibility

After Hurricane Flora, which resulted in some 1,200 deaths, the Cuban Government decided to create of a civil defence structure and to develop the meteorological service that in 1963 only comprised two specialist meteorologists. Today Cuba's response to hurricanes is based on a highly professional and effective meteorological service and warning systems (Lezcano, 1995; Sims and Vogelmann, 2002), and on efforts that focus on mass education, awareness, preparedness and warnings that allow all members of the population to know how to respond when they receive advance notice of a tropical storm (Aguirre and Trainor, 2010).

The Cuban hurricane forecasting and advisory system are similar to the one used by the National Hurricane Center (NHC) in the USA. The Cuban Institute of Meteorology has eight radars, and operational access to satellite pictures (Naranjo Diaz, 2003; Rubeiera, 2012), and also monitors the NHC as well. Operational forecasts are supported by Cuba's own hurricane prediction methods. Cuba is reported to be one of the few western hemisphere countries carrying out significant scientific research into hurricanes (Naranjo Diaz, 2003). The Cuban centre releases severe weather advisories every 12, 6, or 3 hours to correspond to the extent of the hazard (Naranjo Diaz, 2003)

The Institute of Meteorology in Cuba was not always a respected institution. In the 1970s and 1980s, meteorological forecasts in Cuba were jokingly referred to as coming from the "Instituto de Mentirologia," or the Institute for the Study of Lies (as opposed to the Institute for the Study of Weather[5]) (Schuett and Silkwood, 2008). The inability to forecast weather accurately led to low levels of credibility with the Cuban people. Ongoing improvements, which began in the 1980s, included investment in improved forecasting technological, enhanced communication (e.g. simplification of the language used in forecasts) and the replacement of TV broadcasters by trained meteorologists to ensure that weather reports were given accurately (Schuett and Silkwood, 2008). Dr José Rubiera, the Chief Meteorologist of Cuba's Institute of Meteorology, has earned the trust and admiration of the people of Cuba for his knowledge and risk communication skills (Schuett and Silkwood, 2008). This increase in trust coupled with good communication, radio and TV broadcasts of hurricane warnings reach 97% and 96% of the population, respectively (Thompson, 2007). This improved level of trust in Cuba has led to timely and coordinated evacuations, with the evacuation of the most vulnerable communities commencing 72 hours in advance of the anticipated landfall of hurricanes.

### 5.2.2 Disaster response education

In Cuba people are able to understand warning messages and know how to respond appropriately. Disaster preparedness and prevention are part of all school and university curricula (BRI and GRIPS, 2007; Moore et al., 2007; Garret et al., 2007). Every Cuban attends school and the education system plays a major role in creating awareness and preparedness. A wide varieties of teaching methods are used. In addition, every May all adults undergo a mandatory disaster preparedness and response exercise (the "*Meteoro*"), during which evacuation procedures are practised. This simulation exercise allows government officials to identify vulnerable citizens (Keyser and Smith, 2009) and enables the evacuation of the most

---

[5] The Spanish word "meterologia" sounds similar to "mentirologia" which means "lies"





vulnerable communities 72 hours in advance of a storm (Moore et al., 2009). All sectors of society take part from households to businesses to government (Thompson, 2007). In 2007 a new way for disseminating weather information and hurricane forecasts in Cuba commenced: "the Weather Phone". The public has access to a free phone number that provides active advisories and warnings for dangerous weather phenomena (Rubeira, 2012).

The legal framework for present day disaster risk reduction in Cuba was initiated by the Cuban National Civil Defence Act 1976 which issued a mandate that every adult citizen had to undergo civil defence training. In 1997 the legal framework for civil defence was broadened to encompass all aspects related to disaster risk reduction (Thompson, 2007). The 1997 act also detailed the roles of ministries, social organizations, and all public bodies in the case of an emergency. This act also defined centralised decision making with the president, head of civil defence and minister of the armed forces. The law also

sets out decision-making by local authorities when required by the circumstances (Thompson, 2007).

Since 2000 Risk Reduction Management Centres (RRMC) have been implemented in Cuba. A total of 8 provincial and 84 municipal RRMCs, linked to 310 communities have been set up (UNDP, 2015). The objective of the RRMC model is to promote local level decision-making using coordinated early warnings risk and vulnerability studies, together with community preparedness (UNDP, 2015). This model has generated widespread interest in the Caribbean. Commencing in

2009, the lessons learnt by Cuba and capacity building have been offered to other islands at risk in the Caribbean including the British Virgin Islands, the Dominican Republic, Guyana, Jamaica and Trinidad and Tobago (UNDP, 2015).

**5.2.3 Evacuation**

    Mass evacuation is used in Cuba to get the population away from areas at risk from hurricanes (Pichler and Striessnig, 2013). Although evacuations are coordinated by the government they rely heavily on community assistance (Moore et al.,

2009). Neighbourhood Committees for the Defence of the Revolution in Cuba provide information on the location and characteristics of vulnerable people to higher level authorities during emergencies (Wisner, 2009). This shows that social networks are conduits for information flowing from the "bottom up", as well as from the "top down". During an emergency, all available means of transport are mobilized by the local Civil Defence. Abandoned houses are sealed and people's possessions are taken to a safe place (Pichler and Striessnig, 2013). To prevent looting of the evacuated areas they are

patrolled by the military. Community shelters are established in schools and community buildings at the beginning of the 'alarm' stage, where they receive stocks of water, medicines and supplies. The different heads of government ministries cooperate closely. Each shelter has a director, deputy, physician, nurse, psychologist, veterinarian for pets and police and Red Cross representatives (Thompson, 2007; Pichler and Striessnig, 2013).

    In Cuba, the disaster response is overseen by the Integrated Medical Emergency System (SIUM) (Mesa, 2008). Shelters

are prepared 48 hours before the storm is expected and are stocked with food, water, medical supplies and staffed by a doctor, nurse, police and Red Cross representative (Moore et al., 2009). Special attention is paid to planning medical care for vulnerable populations, such as pregnant women, the sick, small children, and the disabled. Mental health counselling is also provided (Mesa, 2008). All doctors are assigned to their neighbourhoods and are able to check on the most vulnerable patients in their homes (Thompson and Gaviria, 2004). Further, teams ensure that reserves of water, food and medications





are already present in areas likely to be isolated in during an event (Mesa, 2008). The SIUM also has a role in hygiene/epidemiological surveillance. For example, following Hurricane Wilma (2005) public health teams, (including hundreds of medical students), conducted house-to-house water quality tests (Mesa, 2008; Deybis Sánchez and Choonara, 2011). In the immediate recovery period, the Cuban government works with international NGOs to ensure safe drinking

water and to repair water systems (Ortiz, 2010). In addition, the Ministry of Public Health issues chlorine tablets to purify water (Thompson and Gaviria, 2004). Finally, hospitals have clearly designated areas for patients during hurricanes with colour-coded directions. These remain marked all year round to avoid confusion during the event (Schuett and Silkwood, 2008).

### 5.2.4 Implications

Hurricane Flora in 1963 acted as a catalyst for creating, what is known in Cuba's as, a "culture of safety" (Pichler and Striessing, 2013). This culture of safety is based on the awareness of procedures to follow in the event of a hurricane; increasing communities' knowledge of stages of emergency warning, where to get information, how to secure houses and where to go for shelter. It also involved building up the basic trust of communities in the government's capacity and intention to protect them. The creation of a culture of safety did not happen overnight and involved a number of measures

that have been implemented over the past 50 years. The effectiveness of the measures is illustrated by the number of fatalities in Cuba from weather-related hazards relative to its neighbours as shown in Fig. 6.

Fig. 7 shows how the different types of resilience have developed over the past 50 years in Cuba following Hurricane Flora, which was a wakeup call for the Cuban government that immediately implemented measures to improve its National Meteorological Service. Compared to Bangladesh, it could be argued that Cuban society transformed more rapidly, with

fatalities from coastal flooding and cyclones declining rapidly since 1963 (see Figs. 5 and 6). However, even 20 years after Hurricane Flora, the trust of people in the veracity of cyclone forecasts was low. Improvements in the way that warning messages were communicated coupled with increases in the accuracy of forecasts helped to increase the public's trust in the technical experts responsible for delivering warnings.

Community organization and a high degree of social capital have also played a key part in increasing resilience to

hurricanes. Cubans are active in several types of social organizations, women organizations, youth organizations, as well as particularly local disaster risk reduction committees. This all act as forums to discuss disaster risk reduction. Cuban communities have consistently increased their social capital to help increase communities' resilience during times of rigorous economic scarcity. This raises the distinct possibility concrete, practical measures to save lives might ultimately depend more on the intangibles of relationships, training, and education than on high cost procedures and resources

(Thompson and Gaviria, 2004;. Pichler and Striessnig, 2013).



## 6. Discussion

In this section, we synthesise the findings of our analysis and present lessons that developed countries can learn from Bangladesh and Cuba about cost effective investments to help improve community resilience to coastal flooding. In both Cuba and Bangladesh the trust between the authorities (at national or sub-national levels), community leaders and civil society (i.e. the population at large) is high. As a consequence evacuation warnings are generally followed. In the USA evacuation zones are delineated to effectively communicate warnings to the public. However, this is often problematic. The evacuation zones need to be communicated through an extensive public awareness campaign (FEMA, 2013). The turnover of emergency management staff is such that the evacuation zones are often forgotten (FEMA, 2013). In some cases evacuations can be politically driven rather than planned, and the evacuation zone terms are confused with floodplain terms (FEMA, 2013).

In the USA, there is no national level decision process concerning evacuation; state and local governments have the authority to make their own decision. Also, there is variable enforcement capacity related to evacuation. Variable adherence to these orders can lead to a failure to evacuate from high-risk areas and unnecessary evacuation from low-risk areas (Wallace et al., 2016). In Cuba, the local government has capacity and the authority to direct the evacuations based on the emergency plan which they and the local community have been responsible for formulating. In Bangladesh, the CPP volunteers work with communities to raise awareness, preparedness and to assist them with evacuations. Engaging members of American communities in the preparation of emergency plans at local level could help increase the trust, cooperation and level of self-help.

In the USA, horizontal evacuation is rightly stressed due to the potentially high level of fatalities that could occur during an extreme coastal flood event in many locations (see Lumbroso et al., 2015; Jonkman, 2007). However, significant number of people do not comply with evacuation orders for various reasons. For example, around 20% of people remained in New Orleans for the landfall of Katrina. There are a range of possible responses to this issue. Following the Bangladesh experience, the provision of vertical evacuation to local, resilient, multipurpose shelters (e.g. schools, libraries, places of worship) warrants further investigation in a US context. The US Department of Homeland Security's STARTM Home Pilot Project is undertaking a pilot project to promote building design that recognises best practices for building resilience to natural hazards. This pilot could be extended to this type of community facilities/shelters. However, there are other approaches to address the issue of non-evacuation and these all need to carefully considered with national, regional and local planners.

The resilience of communities to coastal surges and cyclones in Bangladesh and Cuba has also increased as a result of improvements in the way that warnings and other risk information related to these hazards is communicated. Research recently carried out in the USA showed that most emergency managers and broadcast meteorologists are concerned about the public's lack of comprehension of its vulnerability to storm surges (Morrow and Lazo, 2013 a,b.c). In their opinion, many coastal residents lack sufficient knowledge about their vulnerability and do not comprehend the nature of storm surge





(Morrow et al., 2014). Following hurricane Katrina research indicated that emergency communication plans need to be improved for urban evacuation situations (Brodie et al., 2006). Researchers found that people in low-income areas, need more explicit information on how to reach safety or evacuate if they have no vehicle, financial resources, or place to stay outside the city, or if someone in their family is physically disabled (Brodie et al., 2006).

In the USA, researchers found that many people are "under-concerned" about the risk posed by coastal surges and that their evacuation intent are often over-stated (FEMA, 2013). A post-Katrina behavioural survey found that that most of the respondents in Alabama, Mississippi, and Louisiana could not interpret a National Weather Service storm surge forecast correctly (Gladwin & Morrow, 2006). A survey completed just before Sandy indicated that people expected the primary hazard to be wind (Baker et al. 2012). However, most of the 67 deaths caused directly by Hurricane Sandy in the USA were

due to drowning, primarily when the storm came ashore (C DC, 2012). Interviews conducted with 205 households in Beaufort County, North Carolina one month after Hurricane Irene in 2011 found that two-thirds of the population was unaware that any evacuation order had been issued and a similar percentage did not appreciated the risk posed by coastal flooding (Wallace et al., 2016).

One important way Cuba has increased its resilience to coastal flooding is incorporating disaster risk reduction into the

curricula of the education system at a variety of levels. In the USA, FEMA and Department of Education has been considering strategies to reach children and youth through disaster preparedness in schools and extracurricular activity (Johnson, 2011). However, disaster risk reduction is not required to be included in American schools' curricula at a national level, state and local level. Schools in the USA have the freedom to use and promote disaster preparedness teaching using educational resources of their choosing or can develop their own (Johnson, 2011). More consistency in such education may

be helpful.

During the past few years, FEMA has focused on developing teaching and learning resources for children, including the "FEMA for Kids" website (see https://www.ready.gov/kids). This provides resources for students (aged between five and 18), parents and teachers including lesson plans, classroom activities and interactive games for children. The American Red Cross has developed a curriculum called "Masters of Disaster" to help teachers in the USA integrate important disaster

safety instruction into their regular core subjects such as math, science and social studies (UNISDR, 2007). This is aimed at children aged 5 to 14 and their families with disaster preparedness information, and to promote behaviour change by providing them with the knowledge, skills and tools to effectively prepare for disasters (UNISDR, 2007). US states where the risk of coastal flooding is high could implement school disaster drills. For an initiative on disaster risk reduction in American schools to be sustainable it would require mainstreaming disaster risk reduction into the national school curricula.

The challenge would be for all the relevant government departments to achieve consensus on a plan of action.



## 7. Conclusions

A high level of disaster risk reduction education, social cohesion and solidarity coupled with trust between government authorities, community leaders and vulnerable communities can help to increase communities' resilience to coastal hazards and tropical cyclones. Both Bangladesh and Cuba have, over the past 40 years, made considerable improvements in how the risk posed by cyclones and coastal surges is communicated to isolated and vulnerable communities as well as improving warnings and evacuation/shelters. The trust in all levels in warnings and emergency management is high. However, this has not always been the case in both countries. In the USA, trust in forecasts is high; however, individual and community response to warnings is variable for a number of reasons. One key element is improving the communication of evacuation orders, together with the actual risk posed to people by coastal flooding.

Communities' resilience could be increased by improving communication and the awareness of the risk posed by coastal surges, mainstreaming disaster risk reduction into the education system and building trusted community networks to help isolated and disadvantaged communities improve resilience. To conclude policies that improve risk communication and disaster risk reduction education can have "multiplier effects" that help to increase community resilience to natural hazards.

### Acknowledgements

We would like to acknowledge the support of the US Army Corps of Engineers Climate Preparedness and Resilience Community of Practice who funded this research.

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




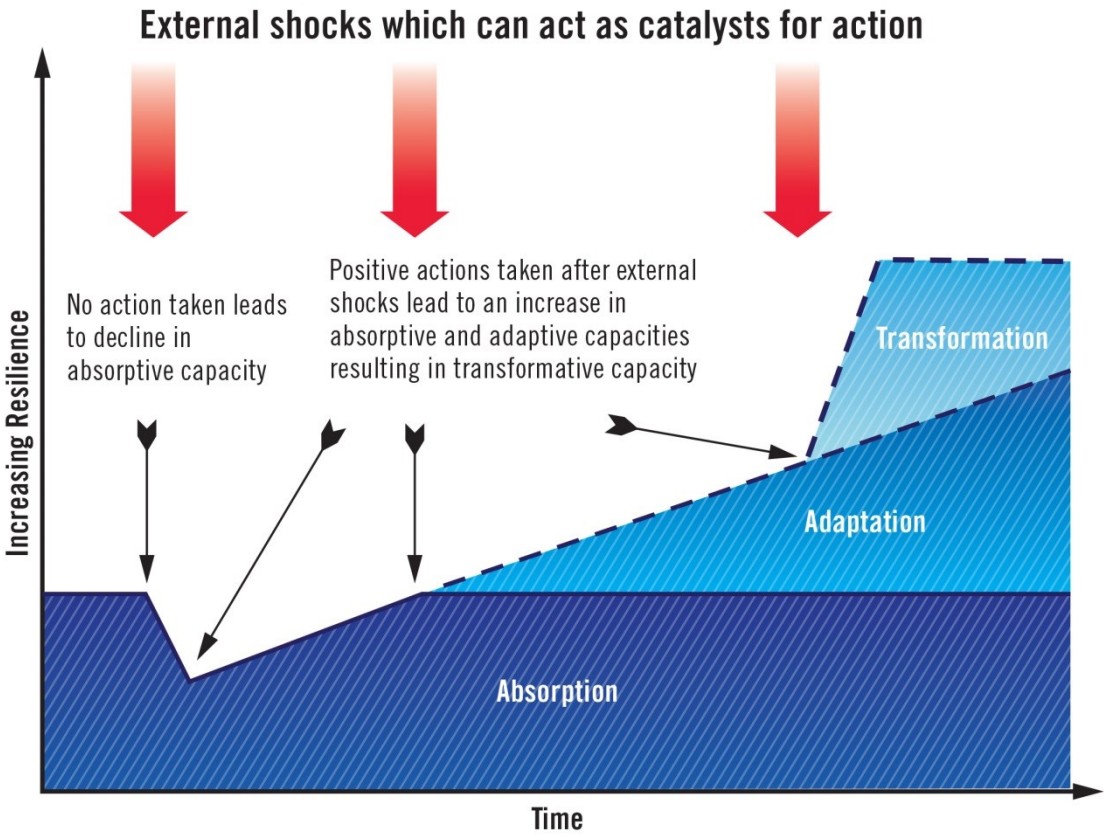

**Figure 1: An illustration of the relationship between adsorption, adaptation, transformation and resilience over time.**




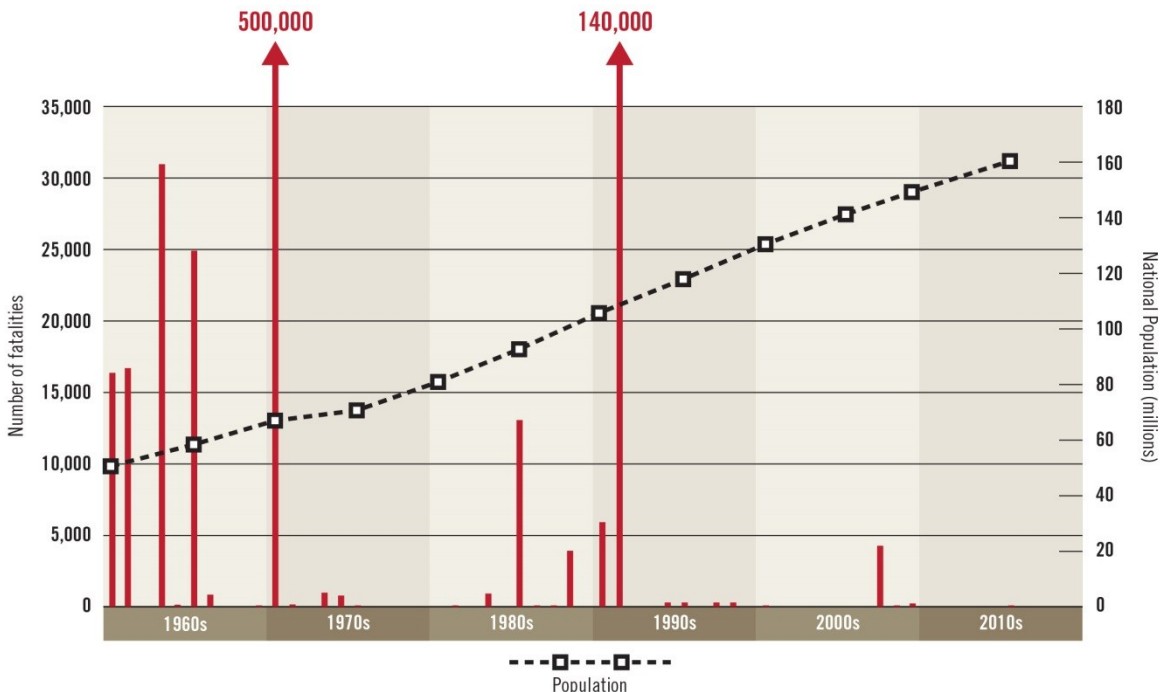

**Figure 2: The number of fatalities caused by tropical cyclones and national population in Bangladesh between 1960 and 2015.**



**Table 1: Inclusion and exclusion criteria**

| Include | Exclude |
|---|---|
| Intervention contributes to resilience | Development action only |
| Intervention focuses on fatalities, health or livelihoods | Does not address coastal flooding |
| Intervention is in response to/anticipation of coastal flooding from a severe storm | Focus on natural system |
| Observable intervention that has been implemented | Not in English |
| | Post-1991 (to include analysis of the period after Cyclone Gorky, Bangladesh) |
| | Incorrect geographic area |
| | Theoretical or speculative (i.e. not observable) |



**Table 2: Documents included/excluded in the review**

| Country | Total peer-reviewed documents | Excluded (and on what grounds) | | Included in review | Total grey literature included in review |
|---|---|---|---|---|---|
| Bangladesh | 1405 | Development only | 413 | 25 | 51 |
| | | Focus on natural systems | 298 | | |
| | | Incorrect geographic scale | 88 | | |
| | | Theoretical or speculative | 456 | | |
| | | Paper not available | 68 | | |
| | | Other | 364 | | |
| Cuba | 138 | Development only | 10 | 6 | 45 |
| | | Focus on natural systems | 67 | | |
| | | Incorrect geographic scale | 16 | | |
| | | Theoretical or speculative | 0 | | |
| | | Paper not available | 6 | | |
| | | Other | 32 | | |





**Figure 3: Timeline for Bangladesh showing the cyclones that have resulted in more than 1,000 deaths, the number of fatalities and implemented risk reduction measures.**



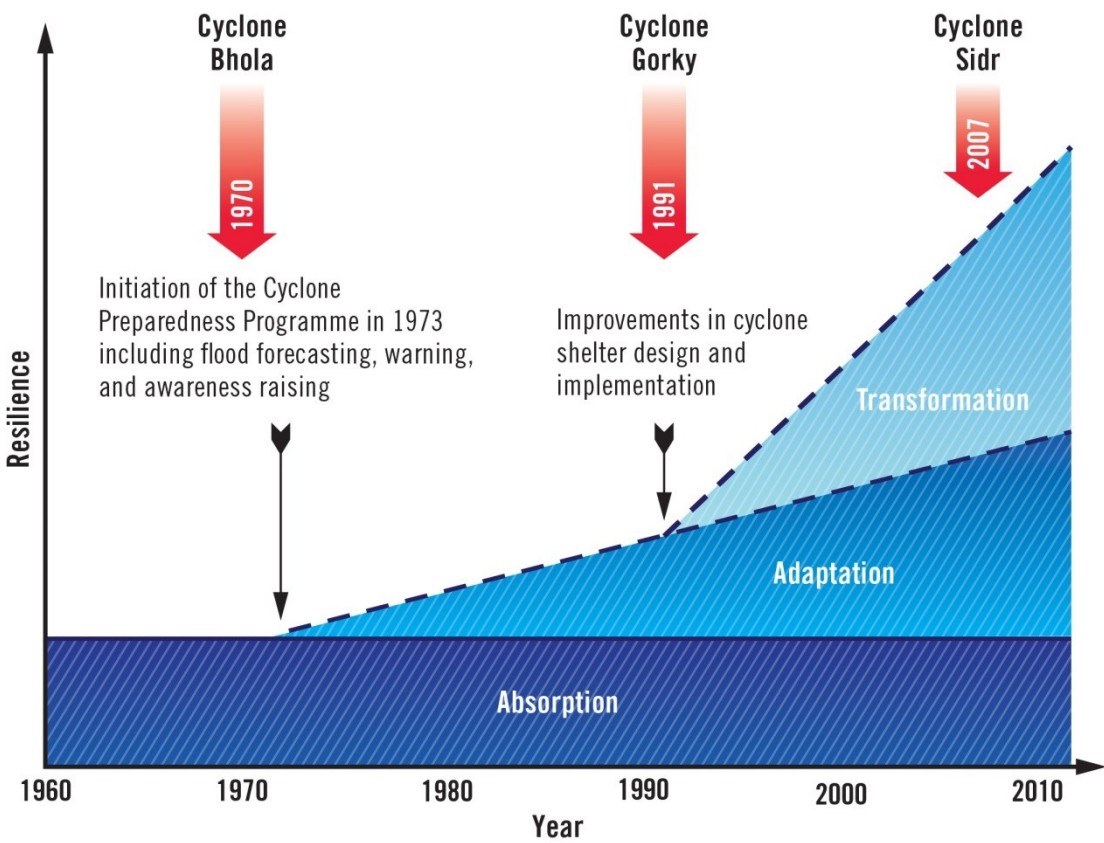

**Figure 4: Development of communities' resilience to cyclones and coastal flooding in Bangladesh over the past 50 years.**





**Figure 5: Timeline for Cuba showing major hurricanes, fatalities attributed to them each decade and implemented risk reduction measures.**





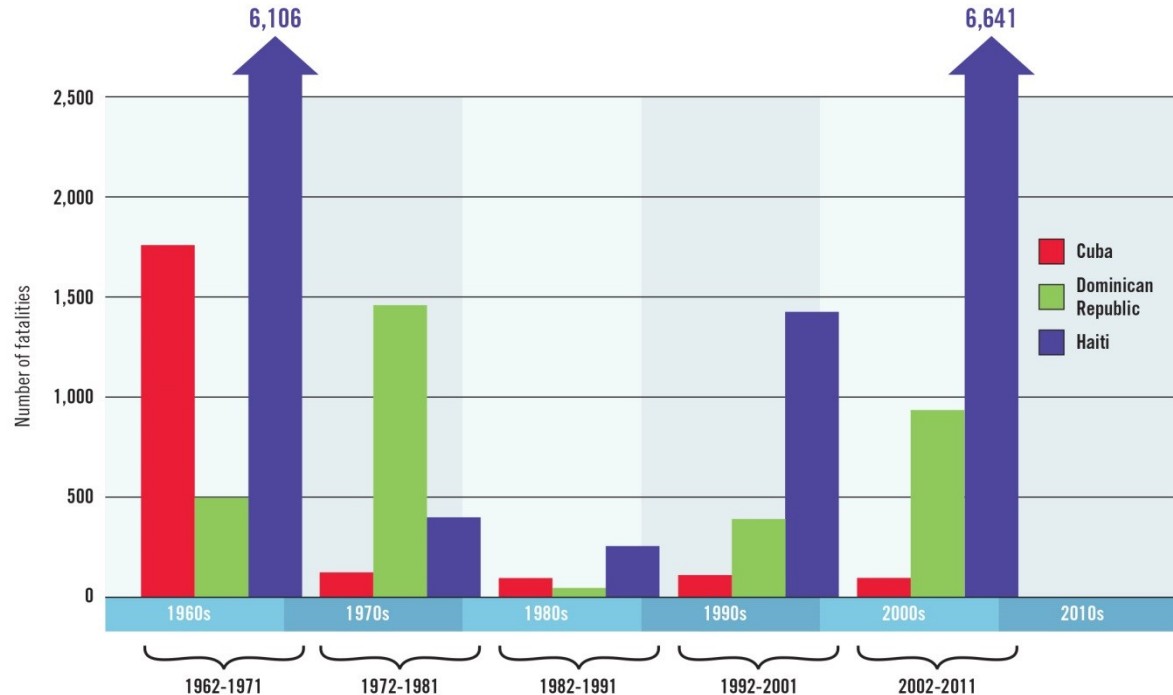

**Figure 6: Total number of deaths per decade from weather-related natural hazards in Cuba, the Dominican Republic, and Haiti between 1962 and 2011.**




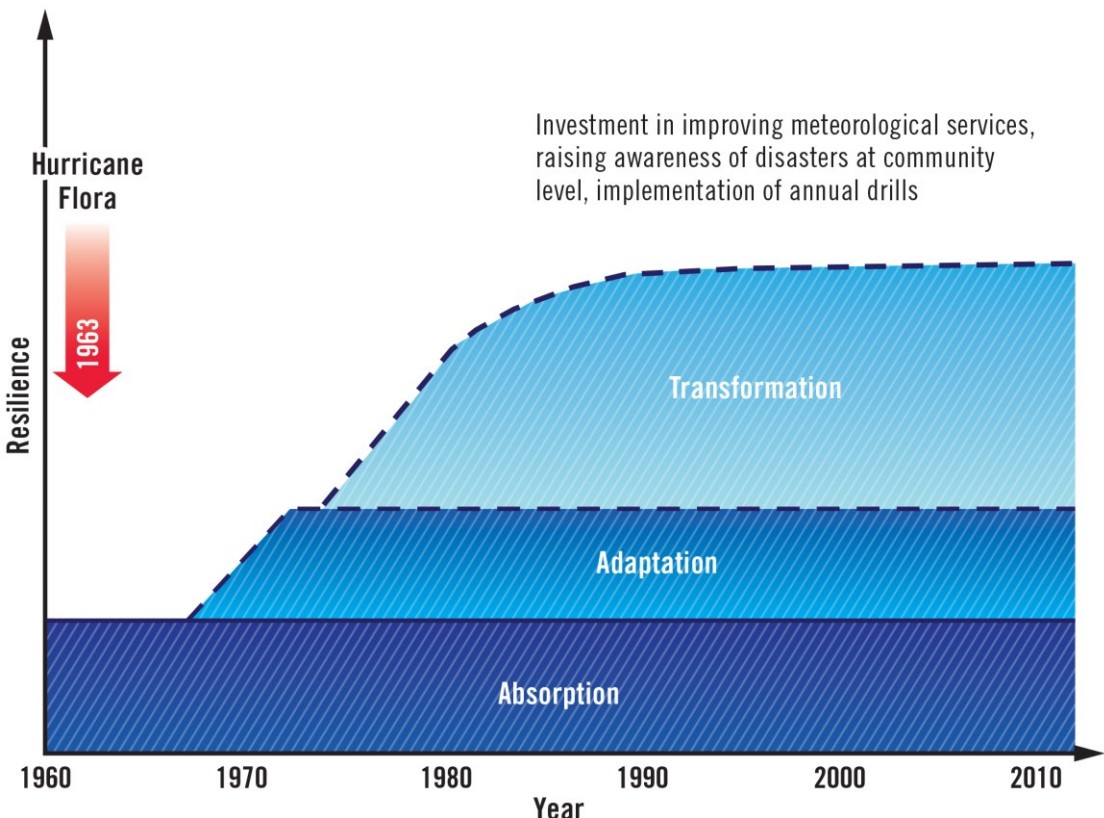

**Figure 7: Development of communities' resilience to cyclones and coastal flooding in Cuba over the past 50 years.**