# Peer review of "Enhancing resilience to coastal flooding from severe storms in the USA: International lessons"

_Natural Hazards and Earth System Sciences, 2016_

## Referee Comment (RC1) · Anonymous Referee #1 · 16 Dec 2016

General comment This is a very interesting paper, well structured and written. The issues addressed are within the scope of NHESS. Important lessons for improving resilience of communities to extreme weather events are provided and conclusions can be very useful to decision-makers. The methodology followed for the systematic review of the peer-reviewed and grey literature related to resilience of Cuban and Bangladesh communities to coastal flooding, is appropriate and well presented. Therefore, the article merits being published, with minor changes.

Specific comments The case of Cuba is clearer to me, in what concerns the effect of the measures implemented on the decreasing number of fatalities. Figure 5 shows the average number per decade, all of which had 2 to 4 major events. One question is

where does the number in the 80s come from, since no events appear in the upper part of the column? Could you clarify this? The case of Bangladesh, though, confuses me. Specifically, in Figure 3 the number of cyclones per decade in the upper part is not always the same with the number of red cubes in the respective 'fatalities' part. Therefore, I think the effect of the implemented actions cannot be easily assessed. You may consider revising this Figure.

Technical corrections: P6, l7: delete the question mark. P6, l24: is it '43 documents' or 127 (Table 2)? P7, l16: I suppose you mean 50,000. P9, l16: I think the use of comma instead of ';' is more appropriate. P9, l20: correct 'warming'. P14, l27: add the verb 'be' before 'carefully considered' P15, l6: delete the second 'that'.

---

## Short Comment (SC1) · 10 Jan 2017

This is a multi-author comment by: C Balk, S Chatterjee, Y Chen, DV Chng, D Colbus, H Davidson, A Harden, J Hilburg, E Jung, C Lim, J Lin, B Marconi, B Moland, J Norris, C Phillips, L Scherr, A Stevenson, J Varamo, C Villar Leeman, A Villegas, J Xu, S Yan, Y Yurong, J Yu, Z Zhou

General comments:

Great study; very beneficial. This paper could be used to help project managers or government branches developing climate-related disaster plans. While many countries have established disaster response plans, educating and simulating these plans could

help make them more successful and highlight vulnerable communities. Many of the issues brought up in the article are issues that people and governments are aware of; however, these precautionary measures can help make plans much more thorough, regardless of existing government policy.

This helps highlight that even developed countries struggle to fully accommodate all the complexities associated with a localized event. Comparing several different approaches between developed and developing countries helps to expand knowledge of potential impacts. For example, Bangladesh has very comprehensive shelters; it is interesting that space for livestock is included in the emergency shelters! While certain plans are more successful and thorough than others, the comparisons help to highlight areas of weakness and strength. We liked how you intrinsically linked preparedness to risk exposure.

Broadly, throughout the paper, the concept of trust is described as if it can be directly measured - how? Trust between communities and different levels of the government is crucial in dealing with flooding hazards. But it is rather difficult to measure the level of trust. What mechanisms can be utilized to effectively measure it? At what levels do you regard it as sufficient to motivate actions against a natural hazard? In addition, the paper mentions the use of social capital in increasing resilience. It seems that social trust is part of social capital. Does measuring trust relate to measuring social capital?

The paper could move more quickly past introductory concepts. Focusing on the methodology and results of study would serve the paper well, in order to make it more concise. The discussion section seems to be the main focus of the paper. We suggest considering that this section could be the "Results" section, and the "Discussion" section would further discuss the results.

We recommend to reduce the backstory in favour of more content (pg 11, line 14). The authors do not seem to be considering the difference in the governmental structures of these three drastically different countries, and do not acknowledge the difficulties in

replicating the same level of response. The political climate of these countries (Cuba, Bangladesh) should be addressed to examine reasons their policies and management are successful or how they are not successful.

The discussion would benefit from more emphasis on socioeconomic implications in the United States. Just because this worked in Cuba and Bangladesh, does not mean it will work in the U.S. These countries are very different from the U.S. in terms of development and culture. The article begs the question, how is American social isolation contributing to the vulnerability and lack of resilience, and how are Cuba and Bangladesh succeeding in this department. Why is the U.S. lacking?

Specific comments:

Table 2 from page 27: Please list the reasons why you excluded certain papers in greater detail. Why did you not attempt to include a wider range of papers in their reviews?

It would be helpful to include some of the same charts for the USA. For example with Figure 5, we have this data for Cuba, what are these data for the USA?

For people not familiar with social capital, a definition might be helpful.

The catalyst of hurricanes is mentioned in the response and reaction in Cuba and Bangladesh. Why did the United States not see a catalyzing effect after major hurricanes such as Katrina?

The focus on the economic budget of the nations could be further clarified. Are you claiming that because Cuba and Bangladesh have a smaller budget and therefore are more creative with how to allocate and prepare for events; is the US not able to be as creative?

The paper seems focused on the functions of a centralized government that has extensive influence. Could you reflect on how these learnings could be applied to a different situation?

Have you looked into technology based strategies? Can the U.S. use technology in their emergency evacuation plans? Similarly, it would be of interest to compare the current communication tools of the nations.

Technical corrections:

Pg 15, Line 1, "hurricane" should be capitalized

Pg 13, Line 26, "This all act" should be "These all act...";

Figure 1, despite definitions given, "adsorption" is left undefined (this definition is necessary)

Page 14, line 8: "The turnover of emergency management staff is such that the evacuation zones are often forgotten (FEMA, 2013)." - Why?

Page 6, Line 6-7, "In this review, we aimed to establish what measures have contributed towards increasing community' resilience in Bangladesh and Cuba?" This should not be a question.

Page 7, Line 5, there should be a space in the heading and number.

Page 14, Line 20 & Page 15, Line 22 - These citations should be standardized. No need for the word "see."

---

## Referee Comment (RC2) · Anonymous Referee #2 · 28 Apr 2017

This paper deals with a very important subject associated with natural hasard, in this case coastal flooding. Understanding resilience of society exposed to such natural hasard remains an open and key question, in particular to anticipate global changes impact in such vulnerable regions.

This paper is really interesting, very well writting. The methodology is clear and very inspiring. I would like to congratualte the authors, because it is not so often that we have the opportunity to read such "novel scientific story".

I would recommand to the authors few modifications to make their figures clearer.

1/ Whenever the "Number of fatalities" is given in the y-axis (Figs 2, 3, 5 and 6), it would

be more informative to use "Number of fatalities / Cyclone".

2/ It is not clear for me how Figs 4 and 7 are proposed based on the text. In Fig. 4, how the slopes of adaptation and transformation, respectively, are estimated ? In Fig 7, the time overlapping of Adaptation and Transformation is not so clear as well as the larger thickness for the Transformation.

Again congratulation for this paper and thank you for this inspiring work.

---

## Author Comment (AC4) · 5 Jun 2017

A PDF version of the paper has been uploaded to allow the referee to see where changes have been made to the paper.

Please also note the supplement to this comment: http://www.nat-hazards-earth-syst-sci-discuss.net/nhess-2016-363/nhess-2016-363-AC4-supplement.pdf

---

## Author Comment (AC5) · 5 Jun 2017

[revised manuscript text omitted]

**Major cyclones to hit Bangladesh causing more than 1,000 deaths**

[Figure]

| 1960s | 1970s | 1980s | 1990s | 2000s | 2010s |
|-------|-------|-------|-------|-------|-------|
| **1960s**
• One hundred cyclone shelters built | **1970 Cyclone**
**1970**
• Agreement on constructing cyclone shelters reached. World Bank funds construction of some 200 shelters | **1980**
• National Forecast Centre of the Institute of Meteorology adopted a user oriented philosophy | **1990 Cyclone**
**1991 Cyclone**
**1991**
• Multipurpose Cyclone Shelters Program commissioned in 1991. Almost 2,000 cyclone shelters constructed | **2000**
• Number of CPP volunteers reaches 32,000 | **2012**
• Number of CPP volunteers reaches 50,000 |
| **1960 Cyclone**
**1961 Cyclone**
**1962 Cyclone**
**1963 Cyclone** | | **1985 Cyclone**
**1985**
• Standing Orders for Cyclone (SOC) proclaimed by the Government of Bangladesh constitute the basic plan for coping with cyclone disasters | **1997**
• Comprehensive Community-Based Disaster Preparedness Programme | **2004**
• Radar systems on coast fail irreparably

**2005**
• Ministry of Food and Disaster Management established | |
| **1964**
• National Water Plan commenced to produce a 20 year masterplan that includes flood defences | **1972**
• Cyclone Preparedness Programme (CPP) of Bangladesh Red Crescent Society set up | | | **2007 Cyclone**
**2007**
• 1.5 million people take refuge in cyclone shelters during Cyclone Sidr | |
| **1965 Cyclone**
**1965**
• International Federation of Red Cross and Red Crescent Societies request the setting up of a warning system for coastal communities | **1973 Cyclone**
**1973**
• Bangladesh Government takes ownership of CPP | **1988 Cyclone**
**1988**
• Two S band radars implemented on the coast for cyclone forecasting installed | | **2009**
• Survey indicates that of the cyclone shelters in Bangladesh 2,583 are usable or moderate usable, 246 are unusable, and 88 have been destroyed

• Under Second Primary Education Programme 924 new cyclone shelters planned | |
| **1968 Cyclone** | | | | **Late 2000s to 2010s**
• Upgrading of coastal radar | |

**Date Cyclone** = Cyclone which caused more than 1,000 deaths

[Figure]

**Figure 3: Timeline for Bangladesh showing the cyclones that have resulted in more than 1,000 deaths, the number of fatalities and implemented risk reduction measures.**

**Major hurricanes to hit Cuba**

[Figure]

| 1960s | 1970s | 1980s | 1990s | 2000s | 2010s |
|---|---|---|---|---|---|
| **1960: Donna** | **1970: Celia** | **1980**
• National Forecast Centre of the Institute of Meteorology adopted a user oriented philosophy | **1991: Elena** | **2001: Michelle** | **2012: Sandy** |
| **1963: Flora** | **1970**
• WMO approves request to extend and improve Meteorological Services | **1981**
• TV weather reports commence being presented by meteorologists | **1994**
• People's Defense Structure transformed into the Civil Defense System with the main objective to safeguard the Cuban population | **2002: Isidore, Lilli** | **2015: Erika** |
| **1963**
• System of civil works to prevent frequent coastal flooding implemented
• Improvements to National Meteorological Service commenced | **1974**
• East Germany donates tracking equipment that allows images in visible and infrared spectrum produced by NOAA to be used to track atmospheric phenomena | **1985: Kate** | **1997**
• Legal framework for civil defense broadened to all aspects related to disaster risk reduction | **2005: Katrina** | |
| **1964: Hilda** | | **1986**
• Annual two day hurricane training exercise for all adults and emergency responders implemented | | **2005 to 2008**
• Setting up of regional disaster risk reduction management centres | |
| **1965**
• Institute of Meteorology and School of Meteorology opened | **1975: Eloise** | | **1998: Georges** | **2007**
• Free phone based weather warning system set up | |
| **1966: Inez** | **1976**
• Mandate issued for all adults to receive civil defense training | | **1999**
• Institute of Meteorology starts forecasting hurricanes with a 90% accuracy | **2008: Ike** | |
| **1966**
• People's Defense structure approved | **1977**
• Joint Cuban-Soviet laboratory for the study of tropical meteorology and hurricanes set up | | | | |
| **1969**
• Weather satellite image receiving station set up | **1979: Frederic, David** | | | | |

Fatalities per decade

2,000
1,500
1,000
500
0

| 1960s | 1970s | 1980s | 1990s | 2000s | 2010s |

**Date: Name** = Major hurricane

[Figure]

Figure 4: Development of communities' resilience to cyclones and coastal flooding in Bangladesh over the past 50 years.

[Figure]

**Figure 54: Timeline for Cuba showing major hurricanes, fatalities attributed to them each decade and implemented risk reduction measures.**

[Figure]

**Figure 65: Total number of deaths per decade from weather-related natural hazards in Cuba, the Dominican Republic, and Haiti between 1962 and 2011.**

[Figure]

**(a)** **Bangladesh**

[Figure]

**(b)** **Cuba**

5 **Figure 6: Conceptual diagram comparing and contrasting the development of communities' resilience to cyclones and coastal flooding in (a) Bangladesh and (b) Cuba since 1960.**

[Figure]

Figure 7: Development of communities' resilience to cyclones and coastal flooding in Cuba over the past 50 years.

Co

Fo

---

## Author Response (AR1)

**Referee #1 – responses**

We wish to thank the referee for their comments and corrections.

With regards to Figure 5 which shows the "Timeline for Cuba showing major hurricanes, fatalities attributed to them each decade and implemented risk reduction measures" we have carried out further research and have found that the majority of fatalities in the 1980s in Cuba were as a result of Hurricane Kate which occurred in 1985.  Figure 5 has been updated and Hurricane Kate has been added to the timeline.

With regards to Figure 3 that shows the "Timeline for Bangladesh showing the cyclones that have resulted in more than 1,000 deaths, the number of fatalities and implemented risk reduction measures" we have revised this figure. An error in the drafting of this figure meant that one major cyclone that occurred in 1963 had been omitted and also that some of the red dots indicating the number of fatalities had been incorrectly positioned.  This figure has now been rectified.

Technical corrections – responses

| | |
|---|---|
| Page 6 Line 7 | The question mark has been deleted |
| Page 6 Line 24 | It is 127 documents.  This been corrected |
| Page 7 Line 16 | This should be 50,000 |
| Page 9 Line 16 | This has been changed from a semi-colon to a comma |
| Pale 9 Line 20 | "Warming" has been corrected to "Warning" |
| Page 14 Line 27 | "be" has been added before the words "carefully considered" |
| Page 15 Line 6 | The second "that" has been deleted |

**Referee #2 – responses**

We would like to thank the referee for their comments and we are pleased that they found the paper interesting.

With regards to the figures we would like to keep the y axis as "Number of fatalities", this is because for one of the figures the y axis is "Number of fatalities for weather-related natural hazards" and hence just using the words "Number of fatalities" helps to avoid any confusion.

With regards to Figs 4 and 7, these are conceptual diagrams showing how the three types of resilience have changed over the years in both Bangladesh and Cuba. We have moved these figures to the discussion and put these them together as Figs 6a and 6b which compare and contrast the development of communities' resilience to cyclones and coastal flooding in (a) Bangladesh and (b) Cuba since 1960.  We have also added some extra text so the cases of how resilience in Bangladesh and Cuba has changed can be contrasted more easily.

**Comments received from Coughlan et al. – responses**

We would like to thanks Coughlan et al. for their collated comments.

The concept of trust is mentioned several times in the paper. We recognise that trust is a difficult concept to measure. In fact it can be argued that trust escapes a simple measurement because its meaning is too subjective for universally reliable metrics. Where attempts have been made to measures trust (see the paper Glaeser et al., 2000 Measuring trust, The Quarterly Journal of Economics) these have been carried out via quantitative surveys of people's perceived levels of trust in various things. Where we have stated in the paper that "trust is high", it is generally based on quantitative surveys carried out by others (e.g. Paul and Rahman, 2006; Roy, 2016 in Bangladesh) or from observations such as those by Schuett and Silkwood, 2008 in Cuba. The question as is posed by Coughlan et al. as to whether measuring trust relates to measuring social capital is an interesting one. Economists have tried to identify the impact of social capital by using attitudinal measures of trust from survey questionnaires. In 1997 Knack & Keefer showed that an increase of one standard deviation in country-level trust predicted an increase in economic growth of more than one-half of a standard deviation (see Knack & Keefer, 1997, Does social capital have an economy payoff ? A cross-country investigation, Quarterly Journal of Economics, pp1251–1288). La Porta et al. also found a correlation between an increase in trust and a decrease in government corruption (see La Portaet et al., 1997 Trust in large organizations, American Economic Review Papers and Proceedings, pp333–338). Although of interest such a quantitative investigation into trust was outside the scope of this research, although we believe it could easily warrant another paper.

With regards to the length of the backstory and introductory comments in light of the positive comments made by Referees 1 and 2 we would like to maintain the introduction as it currently stands as many of these facts are not widely known especially to readers outside North America.

The socio-economic and political systems of both Bangladesh and Cuba are different to the USA. The paper has been modified to bring this point out better. Evidence has also been added as to how lessons from Bangladesh and Cuba have been transferred to other countries with different political systems.

Specific comments – responses

With regards to Table 2 on Page 27 we used a systematic review process to analyse the literature. It is detailed in Table 1 of the paper on what basis papers where or where not included. A wide range of papers and literature was reviewed and it was carried out in a systematic review using methods outlined in various papers that we have cited.

The reason that some of the diagrams produced for Bangladesh and Cuba were not produced for the USA is that the idea of the paper is to show that the various measures that have been introduced in these countries have been successful. It was not felt that adding similar diagrams to the paper for the USA would improve its "readability".

A definition of social capital is given in the footnote on page 3 of the paper.

Regarding the point made regarding the catalysing effect of hurricanes in Bangladesh and Cuba and the question posed as to why the United States did not see one following Hurricane Katrina? We would argue that Katrina (and also Sandy) did act as a catalyst for change and that this work, which was funded by the US Army Corps of Engineers, is part of that! There has been a portfolio of structural

and non-structural measures implemented or being planned in both New Orleans and New York following Hurricane Sandy so there is evidence to suggest these events in the US had a catalysing effect.

5   With regards to economic budgets the purpose of this is to provide an evidence base that both Bangladesh and Cuba are both low income countries. It is not a matter of creativity it is a matter of showing that there are methods that can be used in poor and isolated communities, no matter where they are located in the world, that can be used to increase their resilience.

10  We agree that both Bangladesh and Cuba have centralised government structure.  However, the US has FEMA which is a centralised agency.  We have modified the paper to show that lessons from Bangladesh and Cuba are transferable.

    With regards to technology based strategies this was not the focus of the paper.  The US already uses
15  technology based strategies in terms of forecasting hurricanes and planning for evacuations to name just two examples.  We agree that it would be very interesting to compare the different communication tools used; however, this would potential need to be covered by another separate paper.

    Technical corrections – responses
20
    | Page 15 Line 1 | The word hurricane has been capitalised |
    | Page 13 Line 26 | "This all act" has been corrected to "These all act" |
    | Figure 1 | "adsorption" is a typo this has been corrected to "absorption" |
    | Page 6 Line 6 to 7 | The question mark has been deleted |
25  | Page 14 Line 8 | One of the reasons given for high staff turnover in FEMA in many US Government documents is "low staff morale".  This has been added to the paper. |
    | Page 7 Line 5 | A space has been added |
    | Page 14 Line 20 | This citation has been standardised |
    | Page 15 Line 22 | This citation has been standardised |

[revised manuscript text omitted]

**Major cyclones to hit Bangladesh causing more than 1,000 deaths**

[Figure]

| 1960s | 1970s | 1980s | 1990s | 2000s | 2010s |
|-------|-------|-------|-------|-------|-------|

**1960s**
- One hundred cyclone shelters built

**1960 Cyclone**

**1961 Cyclone**

**1962 Cyclone**

**1963 Cyclone**

**1964**
- National Water Plan commenced to produce a 20 year masterplan that includes flood defences

**1965 Cyclone**

**1965**
- International Federation of Red Cross and Red Crescent Societies request the setting up of a warning system for coastal communities

**1968 Cyclone**

**1970 Cyclone**

**1970**
- Agreement on constructing cyclone shelters reached. World Bank funds construction of some 200 shelters

**1972**
- Cyclone Preparedness Programme (CPP) of Bangladesh Red Crescent Society set up

**1973 Cyclone**

**1973**
- Bangladesh Government takes ownership of CPP

**1980**
- National Forecast Centre of the Institute of Meteorology adopted a user oriented philosophy

**1985 Cyclone**

**1985**
- Standing Orders for Cyclone (SOC) proclaimed by the Government of Bangladesh constitute the basic plan for coping with cyclone disasters

**1988 Cyclone**

**1988**
- Two S band radars implemented on the coast for cyclone forecasting installed

**1990 Cyclone**

**1991 Cyclone**

**1991**
- Multipurpose Cyclone Shelters Program commissioned in 1991. Almost 2,000 cyclone shelters constructed

**1997**
- Comprehensive Community-Based Disaster Preparedness Programme

**2000**
- Number of CPP volunteers reaches 32,000

**2004**
- Radar systems on coast fail irreparably

**2005**
- Ministry of Food and Disaster Management established

**2007 Cyclone**

**2007**
- 1.5 million people take refuge in cyclone shelters during Cyclone Sidr

**2009**
- Survey indicates that of the cyclone shelters in Bangladesh 2,583 are usable or moderate usable, 246 are unusable, and 88 have been destroyed
- Under Second Primary Education Programme 924 new cyclone shelters planned

**Late 2000s to 2010s**
- Upgrading of coastal radar

**2012**
- Number of CPP volunteers reaches 50,000

Fatalities per year

| 1960s | 1970s | 1980s | 1990s | 2000s | 2010s |
|-------|-------|-------|-------|-------|-------|

**Date Cyclone** = Cyclone which caused more than 1,000 deaths

[Figure]

**Figure 3: Timeline for Bangladesh showing the cyclones that have resulted in more than 1,000 deaths, the number of fatalities and implemented risk reduction measures.**

**Major hurricanes to hit Cuba**

[Figure]

| 1960s | 1970s | 1980s | 1990s | 2000s | 2010s |
|---|---|---|---|---|---|
| **1960: Donna** | **1970: Celia** | **1980** | **1991: Elena** | **2001: Michelle** | **2012: Sandy** |
| **1963: Flora** | **1970** | • National Forecast Centre of the Institute of Meteorology adopted a user oriented philosophy | **1994** | **2002: Isidore, Lilli** | **2015: Erika** |
| **1963** | • WMO approves request to extend and improve Meteorological Services | | • People's Defense Structure transformed into the Civil Defense System with the main objective to safeguard the Cuban population | **2005: Katrina** | |
| • System of civil works to prevent frequent coastal flooding implemented | **1974** | **1981** | | **2005 to 2008** | |
| • Improvements to National Meteorological Service commenced | • East Germany donates tracking equipment that allows images in visible and infrared spectrum produced by NOAA to be used to track atmospheric phenomena | • TV weather reports commence being presented by meteorologists | **1997** | • Setting up of regional disaster risk reduction management centres | |
| **1964: Hilda** | | **1985: Kate** | • Legal framework for civil defense broadened to all aspects related to disaster risk reduction | **2007** | |
| **1965** | | **1986** | | • Free phone based weather warning system set up | |
| • Institute of Meteorology and School of Meteorology opened | | • Annual two day hurricane training exercise for all adults and emergency responders implemented | **1998: Georges** | **2008: Ike** | |
| **1966: Inez** | **1975: Eloise** | | **1999** | | |
| **1966** | **1976** | | • Institute of Meteorology starts forecasting hurricanes with a 90% accuracy | | |
| • People's Defense structure approved | • Mandate issued for all adults to receive civil defense training | | | | |
| **1969** | **1977** | | | | |
| • Weather satellite image receiving station set up | • Joint Cuban-Soviet laboratory for the study of tropical meteorology and hurricanes set up | | | | |
| | **1979: Frederic, David** | | | | |

Fatalities per decade

| | 2,000 | 1,500 | 1,000 | 500 | 0 |

| 1960s | 1970s | 1980s | 1990s | 2000s | 2010s |

**Date: Name** ▬ = Major hurricane

[Figure]

Figure 4: Development of communities' resilience to cyclones and coastal flooding in Bangladesh over the past 50 years.

[Figure]

**Figure 54: Timeline for Cuba showing major hurricanes, fatalities attributed to them each decade and implemented risk reduction measures.**

[Figure]

**Figure 65: Total number of deaths per decade from weather-related natural hazards in Cuba, the Dominican Republic, and Haiti between 1962 and 2011.**

[Figure]

[Figure]

5    **Figure 6: Conceptual diagram comparing and contrasting the development of communities' resilience to cyclones and coastal flooding in (a) Bangladesh and (b) Cuba since 1960**

[Figure]

Figure 7: Development of communities' resilience to cyclones and coastal flooding in Cuba over the past 50 years.

Co

Fo